# Histological and Histomorphometric Evaluation of Implanted Photodynamic Active Biomaterials for Periodontal Bone Regeneration in an Animal Study

**DOI:** 10.3390/ijms24076200

**Published:** 2023-03-24

**Authors:** Bernd Sigusch, Stefan Kranz, Andreas Clemm von Hohenberg, Sabine Wehle, André Guellmar, Dorika Steen, Albrecht Berg, Ute Rabe, Markus Heyder, Markus Reise

**Affiliations:** 1Department of Conservative Dentistry and Periodontology, University Hospitals Jena, An der alten Post 4, 07743 Jena, Germanyvonhohenberg@gmx.at (A.C.v.H.);; 2Biolitec Research GmbH, 07745 Jena, Germany; 3Innovent Technologieentwicklung e.V., 07745 Jena, Germany

**Keywords:** alloplastic bone graft, alveolar bone, bone defects, bone label, bone substitute, meso-tetra(hydroxyphenyl)chlorin, ovine bone model, periodontitis, photodynamic therapy, polyfluorochromes

## Abstract

Recently, our group developed two different polymeric biomaterials with photodynamic antimicrobial surface activity for periodontal bone regeneration. The aim of the present study was to analyze the biocompatibility and osseointegration of these materials in vivo. Two biomaterials based on urethane dimethacrylate (BioM1) and tri-armed oligoester-urethane methacrylate (BioM2) that additionally contained ß-tricalcium phosphate and the photosensitizer mTHPC (meso-tetra(hydroxyphenyl)chlorin) were implanted in non-critical size bone defects in the femur (n = 16) and tibia (n = 8) of eight female domestic sheep. Bone specimens were harvested and histomorphometrically analyzed after 12 months. BioM1 degraded to a lower extent which resulted in a mean remnant square size of 17.4 mm², while 12.2 mm² was estimated for BioM2 (*p* = 0.007). For BioM1, a total percentage of new formed bone by 30.3% was found which was significant higher compared to BioM2 (8.4%, *p* < 0.001). Furthermore, BioM1 was afflicted by significant lower soft tissue formation (3.3%) as compared to BioM2 (29.5%). Additionally, a bone-to-biomaterial ratio of 81.9% was detected for BioM1, while 8.5% was recorded for BioM2. Implantation of BioM2 caused accumulation of inflammatory cells and led to fibrous encapsulation. BioM1 (photosensitizer-armed urethane dimethacrylate) showed favorable regenerative characteristics and can be recommended for further studies.

## 1. Introduction

Periodontitis is an infectious and inflammatory oral disease which is characterized by destruction of the tooth supporting tissue [1,2,3]. Clinically, periodontitis appears in signs of inflammation such as bleeding on probing, formation of periodontal pockets, and increased tooth mobility in the later stages. 

A successful periodontitis treatment is characterized by a reduction in periodontal inflammation signs, a decrease in periodontal pocket depths and in long-term suppression of periodontopathogenic bacterial species. 

After initial anti-infectious therapy, periodontal pockets might still persist which is often associated with the presence of deep intrabony defects [4]. 

In those cases, different surgical and non-surgical regenerative procedures are applied which also involve the use of resorbable or non-resorbable membranes, growth and differentiation factors, enamel matrix proteins and autologous, heterologous bone grafts and/or xenografts [5,6,7,8,9,10,11]. 

Because of the high practicability, alloplastic bone grafts such as bioceramics (e.g., absorbable/non-resorbable hydroxyapatite, beta-tricalcium phosphate), bioglasses, metals, calcium phosphate cements and polymers are also of special interest [12,13,14,15]. 

Unfortunately, none of these materials currently meet all necessary clinical requirements such as providing sufficient mechanical stability paired with efficient osteo-inductive and -conductive properties as well as mechanisms to fight local infections of the implant site. 

In this regard, our group already published data showing that polymers of poly(vinyl butyral-co-vinyl alcohol-co-vinyl acetate), urethane methacrylate and functionalized oligolactones have promising characteristics [16]. Due to their highly adaptable nature, synthetic polymers meet the requirements of many biomedical approaches. This includes mechanisms for regulating mechanical properties, porosity, biodegradation, surface topography, and wettability [17,18]. Furthermore, synthetic polymers provide the necessary mechanical strength while being replaced by newly formed bone [19]. The opportunity to include antimicrobial agents into the polymeric matrix is a further advantage [20].

As there is also a high clinical need for new and innovative antibiotic-free materials, the incorporation of a so-called photosensitizer is one favorable approach [21,22,23]. As already proven by various authors, illumination of photosensitizer-armed materials by light of an appropriate wavelength results in sufficient suppression of local microbes [24,25,26]. Up to now, photodynamic active materials are mainly investigated in their efficiency to treat infected wounds or are tested for their practicability in tumor therapy [27,28,29,30,31]. 

To the best of our knowledge, there are currently no studies available that focus on photosensitizer-armed materials for bone regeneration. To fill this gap, our group recently introduced two different photosensitizer-doted biomaterials based on urethane dimethacrylate (BioM1) and tri-armed oligoester-urethane methacrylate (BioM2) [32]. In order to ensure a sufficient photodynamic antimicrobial effect, the photosensitizer meso-tetra(hydroxyphenyl)chlorin (mTHPC) was included into the matrix of both polymers. As already proven, mTHPC is of strong photodynamic activity and capable in suppressing oral pathogens to significantly high extents [32,33,34]. 

Up to now, osseointegration and biocompatibility of BioM1 and BioM2 were not yet observed in vivo. Therefore, the present animal study aimed to investigate the performance of both materials in non-critical bone defects after implantation for 12 months. Both materials were analyzed by histomorphometric and histological methods in order to determine their osseoinductive and bone integrative characteristics. Furthermore, newly formed tissue and the adjunctive bone were investigated for signs of adverse effects and inflammatory responses. 

## 2. Results

In the present animal study, two different polymeric biomaterials armed with mTHPC were implanted in the femur and tibia of sheep. After 12 months of implantation bone samples were collected and analyzed by histological and histomorphometric methods.

After 12 months, BioM1 remnants showed a mean square area of 17.4 mm², while 12.2 mm² were estimated for BioM2. At baseline an initial area of the bone defect was calculated with 19.6 mm². Remnants of BioM2 were significant smaller in the tibia (*p* < 0.001) and in the femur (*p* = 0.007) as compared to BioM1 (Figure 1a).

Total bone was detected in the ROI by 30.3% in femur defects filled with BioM1, whereas only 8.4% was found in case of BioM2. Both values were also significantly different (*p* < 0.001). In the tibia defect sites a total bone value of 28.6% was estimated in the ROI for holes filled with BioM1 and 20.4% for those obturated with BioM2. The results are shown in Figure 1b.

In addition to the bone volume, the percentage of fibrous soft tissue in the ROI was also determined. BioM1 showed a soft tissue value of 3.3% in the femur samples, while in case of BioM2 a higher soft tissue value was detected (29.5 %, *p* < 0.001). In case of the tibia defects, soft tissue was estimated by 3.8% for BioM1 and 15.8% for BioM2 (*p* = 0.014), (Figure 1c).

In the present study, the bone-to-biomaterial contact was also evaluated at the bone interface. It was found that BioM1 showed a bone-to-biomaterial ratio of 81.9% in the femur and 56.2% in the tibia (*p* = 0.005). In contrary, BioM2 showed a bone-to-biomaterial contact of only 8.5% in the femur and 16.4% in the tibia. The results for both biomaterials are displayed in Figure 1d.

As observed in the decalcified histological bone sections, BioM2 was encapsulated by soft tissue to a much higher extent as compared to BioM1. Additionally, an infiltration with fat and giant cells was only observed in defects filled with BioM2. 

In case of BioM1, a pronounced formation of trabecular bone was observed. Though, only minor bone formation occurred in defects filled with BioM2. Representative decalcified and stained histological sections of BioM1 and BioM2 are shown in Figure 2.

Defects filled with BioM1 showed no signs of inflammation or adverse effects. The interface of implanted BioM1 revealed homogeneous cancellous bone in close contact to the biomaterial surfaces.

On the other hand, remnants of BioM2 were enclosed by a sheath of fibrous tissue. Furthermore, osteolytic zones filled with fibrous tissue and fat cells were also discovered. Adjunctive tissue of BioM2 was affected by a strong infiltration with lymphocytes and giant cells (Figure 3).

The results of the assigned four-graded ROI evaluation score are presented in detail in Figure 4. It is clearly demonstrated that a score of grade 1 (new formed bone, totally mineralized) was detected most frequently in defects filled with BioM1 (*p* = 0.02). In contrast, in defects filled with BioM2 a score of grade 3 (fibrous soft tissue, uncalcified) was primarily assigned (*p* = 0.004). 

In summary, bone defects filled with BioM1 showed high amounts of mineralized bone in the ROI (96.18%), while fibrinous tissue was detected for 3.82%, only. In the case of BioM2, bone was present by 18.3% only, while the amount of uncalcified fibrous tissue was clearly increased (grade 3).

## 3. Discussion

In the present study, osteointegration and biocompatibility of two polymeric biomaterials based on urethane dimethacrylate (BioM1) and a tri-armed oligoester-urethane methacrylate (BioM2) were investigated in an ovine bone model. Both biomaterials were implanted in non-critical size defects in the femur and tibia of sheep.

Osteointegration and biocompatibility was determined by histomorphometric analysis after an implantation period of 12 months. In detail, the remaining biomaterial size, the percentage of bone and soft tissue in the ROI as well as the bone-to-biomaterial contact were evaluated. 

As shown by the results, BioM1 was sufficiently osseointegrated with the highest amount of mineralized tissue in the ROI. Results from the four-graded classification scale showed bone formation by 96.18%. In contrast, implantation of BioM2 (oligoester-urethane methacrylate based) caused chronic inflammation and fibrous encapsulation. In the case of BioM2, bone in the ROI was only detected by 18.3%.

Similar results are reported from scaffolds fabricated from polymethyl methacrylate (PMMA) which were also not osseointegrated after implantation for 12 months. The same as for BioM2, the implanted material was encapsulated by a sheath of fibrous tissue. In comparison, mineralized tissue was found for titanium scaffolds by 39.1%, followed by implants manufactured from poly(D,L-lactic acid) (31.5%) and porous ultra-high molecular weight polyethylene (6.4 to 10.1%) [35]. 

The performance of methacrylate-based grafting was also observed by other authors. Recently, it was reported that bioscaffolds composed of Sr-containing mesoporous bioactive glass nanoparticles embedded in a gelatin methacrylate matrix present enhanced osteogenic, angiogenic, and immunomodulatory properties [36]. Furthermore, a novel graphene oxide modified expandable polymethyl methacrylate-based bone cement revealed improved physiochemical properties with sufficient cytocompatible, and osteogenic characteristics [37]. Methacrylated silk was also recently tested to verify its ability to support osteogenesis. It was shown that scaffolds from methacrylated silk are biocompatible and present reliable osteoconductive features [38]. Moreover, the performance of 3D printed gelatin methacrylate hydrogel has formerly been investigated after implantation in rat condyle defects. Whereas optimal tissue integration was observed via histology, no signs of fibrotic encapsulation or inhibited bone formation were attained [39]. 

Using sheep for biomaterial testing is common, especially in orthopedic research, because their parameters are similar to those of humans, such as the anatomic structure of bone and joints, body weight, mineral bone metabolism and responses to mechanical loads [40,41]. The applied model was first introduced in 2008 and revised in 2014 [42,43]. In the present investigation a modified version was introduced which allows serial sampling in the same animal with similar environmental conditions. If necessary, all stages in bone healing can easily be addressed. The applied surgical procedure was also well tolerated by the experimental animals. Although sheep cancellous bone models are now well established for the assessment of new bone substitutes, the limited availability of cancellous bone makes it difficult to find multiple comparable sites within the same animal [42]. Therefore, the described ovine model was chosen for testing biocompatibility and osseointegration of BioM1 and BioM2 in the present investigation.

In addition to large animal studies, in silico methods are of increasing interest. Computational simulation approaches for investigating mechano-biological principles behind scaffold-guided bone regeneration and the influence of the scaffold design on the regeneration process are already described [44,45]. Especially for the treatment of large bone defects with manufactured bone grafts and in joint replacement surgery, in silico analyzation methods show great predictive potential [46,47].

However, in the present investigation, both materials degraded to different extents. At the end of the study period a mean square area of 17.4 mm² was detected for remaining BioM1 and 12.2 mm² for BioM2, which was statistically significant in respect to the defect size estimated at baseline (19.6 mm²). 

These results are in line with findings observed by our group previously. As shown, BioM2 degrades to a much faster extent compared to BioM1. During immersion in distilled water for 28 d, BioM2 lost weight by 67%, while BioM1 degraded by only 4% [32]. 

The inert nature of BioM1 can thereby be referred to its hydrophobic chemical structure. Unlike BioM2, which is of higher hydrophilicity, BioM1 withstands hydrolytic cleavage to a much greater extent [48].

In the present study, the bone-to-biomaterial contact ratio was analyzed as well. As shown by the results, BioM1 presented a bone-to-biomaterial-contact of 81.9%. In contrast, a contact rate of only 8.5% was observed for BioM2.

Overall, the bone-to-biomaterial contact of BioM2 can be considered rather low. In a similar study, osseointegration of titanium and polyetheretherketone (PEEK) implants in the tibia and femur of sheep were observed. The results revealed a percentage of the bone-to-implant contact by 59.3% for titanium and 11.5% for PEEK [49]. 

In the present investigation, BioM2 was affected by fibrous tissue encapsulation, while BioM1 showed a close contact to the surrounding bone. Similar results as for BioM1 were reported for implants manufactured from hydroxyapatite. Here, a bone-to-implant contact of 74% was reported [50]. In this context, BioM1 showed a mean bone-to-biomaterial ratio of 81.9% in the femur and 56.2% in the tibia.

In addition, no inflammatory reactions or fibrous tissue formation was observed in the ROI of BioM1. In contrast, the implantation of BioM2 resulted in chronic inflammation. The inflammatory response associated to BioM2 can probably be referred to cytotoxic byproducts that origin from the degradation process. As it was shown, hydrolytic cleavage of polyester urethane acrylates causes the emerging of various acidic substances such as poly-(methacrylic acid), ethylene glycol, diethylene glycol, lactic acid and glycolic acid which leads to a local drop in the tissue pH [51]. 

In this regard, it is known that the appearance of acidic degradation products causes tissue inflammation and an impaired healing [52]. In order to increase biocompatibility and to counteract the cytotoxic effects of the acidic degradation products, calcium phosphate particles are often additionally applied to the polymeric matrix [17,52,53]. 

In the present study, both polymers were also additionally substituted with ß-tricalcium phosphate nanoparticles for increasing the porosity of the biomaterial body and to improve osseointegration [32,54]. In this context it was observed that especially tortuosity has a significant effect upon the scaffolds’ permeability and shear stress values [55]. Morphologic parameters such as porosity, specific surface area, thickness, and tortuosity are important and hence need to be discovered for BioM1 and BioM2.

In the case of BioM2, giant cells and osteolytic bone defect zones were discovered in the adjunct tissue. The formation of foreign body giant cells is in general a result of fused macrophages that faced a frustrated process of phagocytosis [56,57,58]. The presence of foreign body giant cells, osteolytic zones and signs of a fibrous encapsulation indicate that BioM2 is of rather low biocompatibility.

As a result of the inflammation process, BioM2 also showed a lower bone-to-biomaterial contact rate as compared to BioM1. In detail, BioM2 featured a bone contact of 8.5% in the femur and 16.4% in the tibia, while implantation of BioM1 resulted in a bone contact of 81.9% in the femur and 56.2% in the tibia. After 12 months of implantation, it was recognized that BioM2 was almost entirely enclosed by a sheath of fibrous tissue, while in case of BioM1 no signs of adverse effects were observed. 

The formation of a fibrotic capsule can be referred to a variety of pro-fibrotic growth factors such as PDGF, VEGF, and TGF-β, which are secreted by macrophages and also by several other immune cells. These factors cause activation of fibroblasts and endothelial cells, which start to deposit collagen and other extracellular matrix proteins on the surface of the grafted material. The deposited matrix subsequently matures into a peripheral fibrous capsule, which causes mechanical impairment and insufficient interactions of the biomaterial with the adjunct tissue [59]. 

Both biomaterials, also contained the photosensitizer mTHPC that enables a strong antibacterial surface effect upon illumination with light at 652 nm. As shown by various authors, antimicrobial photodynamic therapy (aPDT) is efficient in suppressing different oral pathogenic bacterial species to significant high extents [33,34,60,61,62,63]. aPDT is also considered an alternative to the systemic treatment of biofilm-related infectious diseases with antibiotics [64,65,66]. Due to the incorporation of mTHPC into the biomaterial matrix, singlet oxygen and other reactive oxygen species (ROS) are produced upon light exposure causing destruction of adherent bacterial cells. Investigations of our group have already shown that illumination of the mTHPC-doted biomaterials with red laser light (652 nm) caused complete inhibition of *Porphyromonas gingivalis* and led to a significant decrease in *Enterococcus faecalis* [32]. The photodynamic antimicrobial activity of both implanted materials was not observed in the present investigation, which limits the overall merit. However, it still needs to be determined whether the photodynamic activity of both biomaterials is also efficient in vivo. A further limitation might be the fact that no additional controls are included. Therefore, the bone regenerative capacity is not comparable to already known grafting materials. Further, the number of tibial bone defects might be increased in order to obtain an even distribution of samples. Information upon the morphology of BioM1 and BioM2 is still limited. Parameters such as porosity, specific surface area, thickness, and tortuosity are important and need yet to be investigated in detail.

Up to now, various photodynamic active materials are already under investigation [67]. However, examinations upon the efficiency in vivo, especially in the case of periodontal lesions are hence needed. 

## 4. Materials and Methods

### 4.1. Characterization of the Biomaterials

In the present animal study, two light curable biomaterials, BioM1 based on urethane dimethacrylate and BioM2 based on tri-armed oligoester-urethane methacrylate (BioM2) were applied. Both investigated biomaterials, BioM1 and BioM2, additionally contained β-tricalcium phosphate microparticles loaded with 20 wt% of the photosensitizer mTHPC. All chemicals were obtained from Sigma-Aldrich Chemie GmbH, Taufkirchen, Germany. The photosensitizer mTHPC was kindly provided by biolitec research GmbH, Jena, Germany. The biomechanical and antimicrobial photodynamic properties of both materials were evaluated by our group in a previous examination [32]. Structural formulas of the applied polymers (urethane dimethacrylate—BioM1, tri-armed oligoester-urethane methacrylate—BioM2) are presented below (Figure 5 and Figure 6).

### 4.2. Surgical Procedure and Biomaterial Application

All experiments were conducted in accordance to the German law of animal protection and welfare. The investigation was authorized by the Thuringia Regional Office for Food Safety and Consumer Protection (protocol code: 02-036/10; date of approval: 14 October 2010). 

Eight female domestic sheep (*Ovis gmelini aries*) obtained from a local breeder with a mean age of 12 months were used in this prospective study. Prior to surgery, all animals were acclimated for 2 weeks at the Central Animal Facility and Service Department, University Hospitals Jena, Germany. The sheep were assigned into two groups with four animals each. In the first group biomaterial 1 (BioM1) and in the second group biomaterial 2 (BioM2) was implanted. 

For implantation, the femoral (distal) and tibial (proximal) epimethaphysial region of the right hind limb was chosen. Surgery was performed under general anesthesia. The animals were placed in right side recumbency and the skin of surgical side was disinfected with iodine (Braunoderm^®^, B.Braun AG, Melsungen, Germany). At first, an approximately 10 cm long incision was applied at the medial side of the distal femur epiphysis 1 cm proximal of the knee joint capsule longitudinally and parallel to the bone axis. The cortical bone was reached through incision of the local muscles and by dissection of the periosteum. 

Two 5 × 6 mm cylindrical holes were drilled in the femoral epiphysis by using a water-cooled trephine. A minimal distance of 20 mm was kept in between the drilled holes to reduce the risk of fracture and to ensure proper healing and harvesting of the bone specimens at the end of the study. 

The defects were filled with either BioM1 or BioM2. Prior to insertion of the biomaterials, the holes were dried using sterile cotton balls. When relative dryness was reached, the gel-like biomaterials were quickly injected in 2 mm thick layers and instantly photopolymerized for 40 s each using a calibrated dental light curing unit (Bluephase, 830 mW/cm², Ivoclar–Vivadent, Ellwangen, Germany). Polymerzation is shown in Figure 7a. After the bone defects were completely filled, the surface of each polymerized biomaterial was whipped once with a 70% ethanolic solution (Figure 7b). The position of the filled defects sites was marked by insertion of a 4 mm long titanium pin (Geistlich Biomaterials, Baden-Baden, Germany). The position of the pin in relation to the defect sites was transferred to a transparent plastic foil which was used for relocation after euthanasia. Subsequently, the periosteum was closed and the muscle fascia, subcutaneous tissue and skin were sutured with an absorbable thread. 

Afterwards, a second approximately 5 cm long incision was applied at the medial side of the proximal tibial epiphysis 1 cm distal of the knee joint capsule longitudinally and parallel to the bone axis. The cortical bone was exposed as described above and another defect of identical dimension was prepared and obturated by the identical biomaterial as already implanted into the femur. 

After marking the location of the implant sites using a titanium pin and plastic foil, a suture was applied. Finally, both surgical sites were dressed with an aluminum based wound spray and medio-lateral as well as dorso-plantar X-ray images were taken as controls and for documentation of the healing progress. The location and total number of filled defect sites are summarized in Table 1. Antimicrobial prophylaxis and post-surgical pain control were applied. Animals were euthanized after 12 months of biomaterial implantation using a standardized protocol.

### 4.3. Sample Preparation and Histological Sectioning

After euthanasia, collected bone specimens were fixed in 5% formaldehyde solution for 5 d and subsequently cut under constant water cooling in two halves using the LEITZ 1600 microtome (Leica Microsystems GmbH, Bensheim, Germany). Each cortical halve of the bone sample was subjected to dehydration in solutions with increasing content of ethanol (50%, 70%, 80%, 2 × 96%, 3 × 100% ethanol) and afterwards embedded in Technovit 9100^®^ (Kulzer GmbH Kulzer Technik, Wehrheim, Germany). The embedded specimens were then sectioned using the LEITZ 1600 microtome (Leica Microsystems GmbH, Bensheim, Germany). Subsequently, each sample was grounded to 10–20 µm of thickness using abrasive papers with different granulation from 300 to 4000 grit and subjected to Masson-Goldner staining. 

The second halve of the divided bone sample was decalcified in 25% EDTA (pH 7.4) at 37 °C for 4 to 10 weeks. The decalcifying procedure was completed when the specimen could easily be penetrated by a fine needle. 

After the decalcification process, samples were dehydrated in an alcoholic series and embedded into paraffin. Each paraffin block was then sectioned and the obtained 5 µm thick samples stained with hematoxylin eosin (HE). Histological allocation of the collected bone specimens can also be observed in Figure 8.

### 4.4. Histomorphometry

Undecalcified and stained histologic sections (n = 30) were observed using the microscope Jenaval (Carl Zeiss MicroImaging GmbH, Jena, Germany) at 10× to 125× magnification. Microscopic images were documented using the software AxioCam^®^ and AxioVision^®^ (release 4.6.3., Carl Zeiss MicroImaging GmbH, Jena, Germany). Data was analyzed using the freeware ImageJ^®^ (1.50i, Wayne Rasband, National Institutes of Health, Bethesda, MD, USA). An ROI (region of interest) around the implanted biomaterial of 500 µm in width was defined and the percentage of bone and soft tissue determined (Figure 9).

In detail, each histological section was analyzed with regard to the square area of the biomaterial remnants, percentage of bone and soft tissue in the ROI and biomaterial-to-bone contact ratio. 

Decalcified and HE stained sections were observed using the microscope Jenaval (Carl Zeiss MicroImaging GmbH, Jena, Germany) at 1× to 250× magnification. From each bone specimen, five different sections were chosen and evaluated by applying an ROI of 250 µm (Figure 10).

The ROI was examined by applying a four-graded scoring system (Table 2). The score was adapted and modified [68] and comprises elements from the DIN EN ISO 10993-6:2017 guidelines [69]. Examples for each score (1–4) are presented in Figure 11. A workflow of the study is presented in Figure 12.

## 5. Conclusions

The results of the present study revealed that BioM1 (photosensitizer-armed urethane dimethacrylate) was bone-integrated to a significantly higher extent compared to BioM2 (photosensitizer-armed oligoester-urethane methacrylate). In case of BioM1, high-quality bone was formed in the ROI without any signs of adverse effects. Due to the slow degradation of BioM1, structural stability is provided for a longer period of time. In contrast, implantation of BioM2 resulted in chronic inflammation and increased fibrous tissue formation at the bone-to-biomaterial interface. 

It can be concluded that BioM1 has promising regenerative and biocompatible characteristics. The material can therefore be recommended for further studies that focus on bone regeneration in regions where an additional structural support as well as stabilization is needed. Hence, it needs to be investigated, if those materials are capable of treating intrabony periodontal lesions sufficiently. Moreover, detailed information upon the antibacterial efficiency of photosensitizer-armed grafting materials still has to be obtained in vivo.

## Figures and Tables

**Figure 1 ijms-24-06200-f001:**
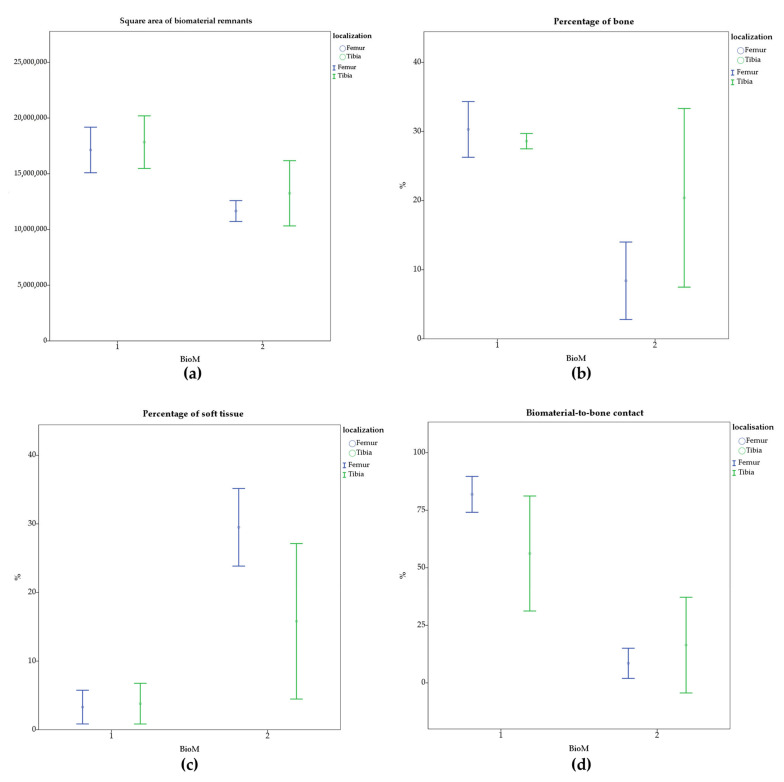
Histological evaluation after implantation of BioM1 and BioM2 in the femur and tibia of sheep for 12 months: (**a**) square area of the biomaterial remnants; (**b**) percentage of bone in the ROI; (**c**) percentage of soft tissue in the ROI; (**d**) bone-to-implant contact at the biomaterial to tissue interface.

**Figure 2 ijms-24-06200-f002:**
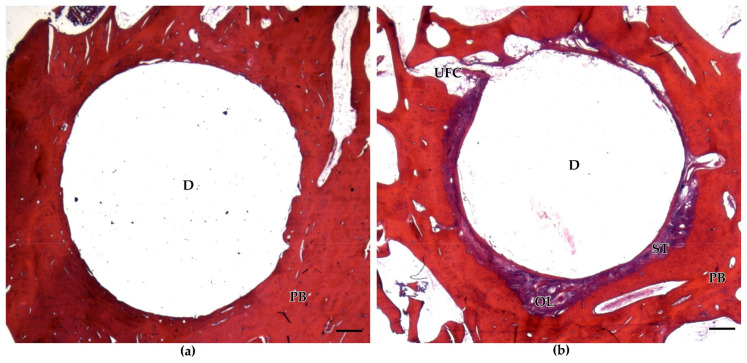
Representative decalcified and stained histological sections of BioM1 and BioM2: (**a**) defects (D) filled with BioM1 revealed a homogeneous bone-to-biomaterial interface with no signs of fibrous encapsulation; (**b**) defects filled with BioM2 are characterized by a pronounced enclosure with fibrous tissue (purple, ST). Further, osteolytic zones (OL) and univacular fat cells (UFC) are additionally present within the bone-to-biomaterial interface of implanted BioM2. Scale bar 500 µm. (PB: peripheral bone).

**Figure 3 ijms-24-06200-f003:**
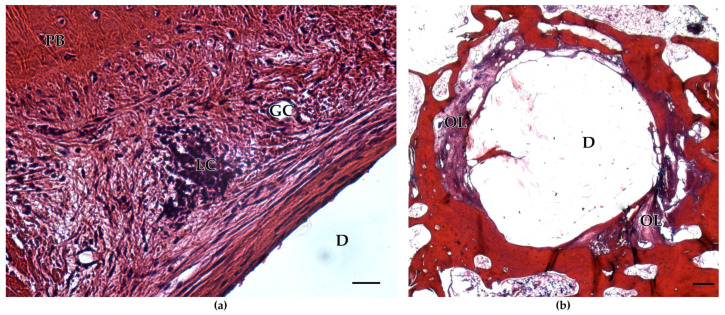
Histological sections of implanted BioM2: (**a**) surrounding tissue shows infiltration by lymphocytes (LC) and giant cells (GC) (left image). Peripheral bone (PB) is present in the left upper corner of the defect site (D). Scale bar 50 µm; (**b**) osteolytic zones (OL) filled with fibrous tissue and inflammatory signs are present in the interface regions. Scale bar 1000 µm.

**Figure 4 ijms-24-06200-f004:**
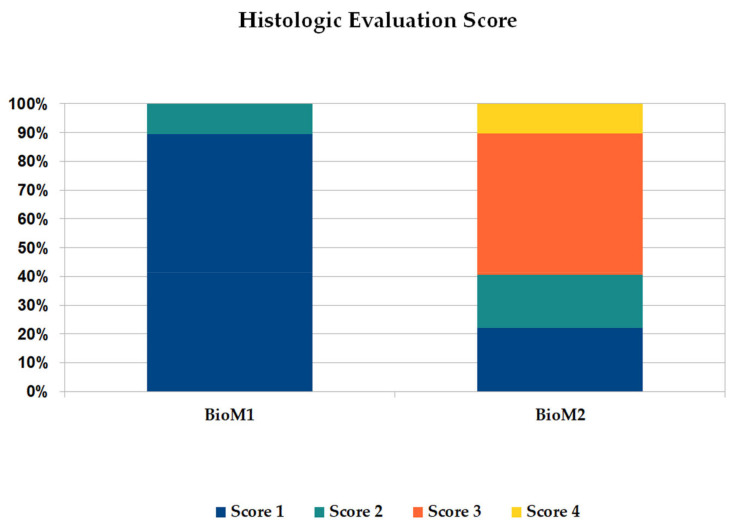
Summary of the four-graded histological evaluation score after implantation of BioM1 and BioM2 for 12 months.

**Figure 5 ijms-24-06200-f005:**
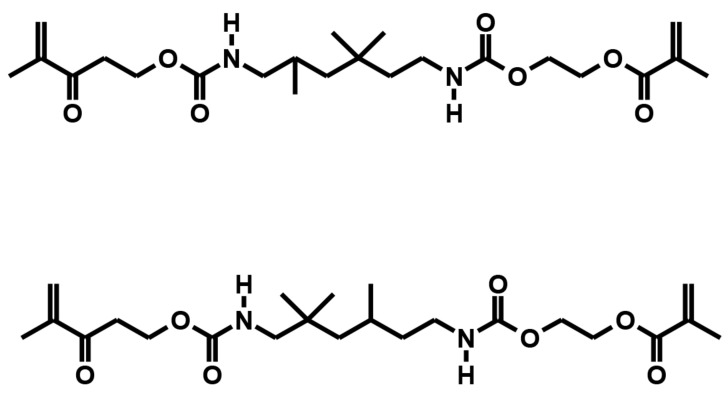
Structural formula of urethane dimethacrylate (the main component of BioM1).

**Figure 6 ijms-24-06200-f006:**
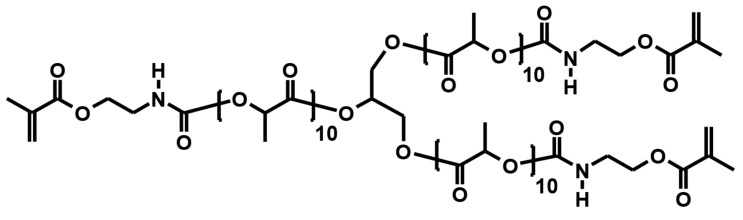
Structural formula of oligoester-urethane methacrylate (the main component of BioM2).

**Figure 7 ijms-24-06200-f007:**
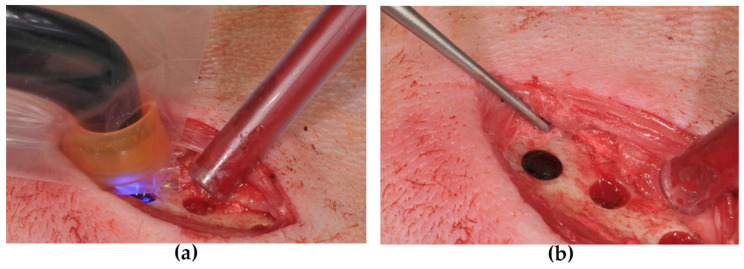
Biomaterial application: (**a**) light-polymerization; (**b**) solid biomaterial in the femur.

**Figure 8 ijms-24-06200-f008:**
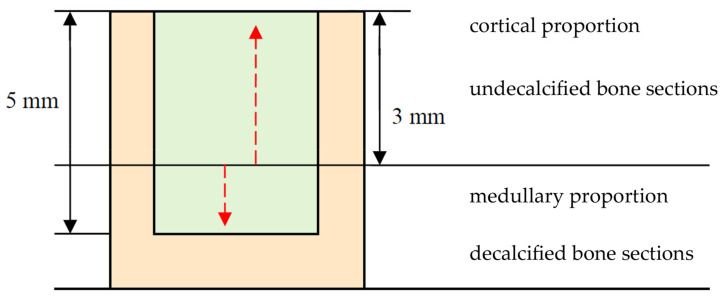
Assignment of the collected bone sample to the respective histologic evaluation method. The biomaterial remnant is expressed by light green. Bone is presented in light red color.

**Figure 9 ijms-24-06200-f009:**
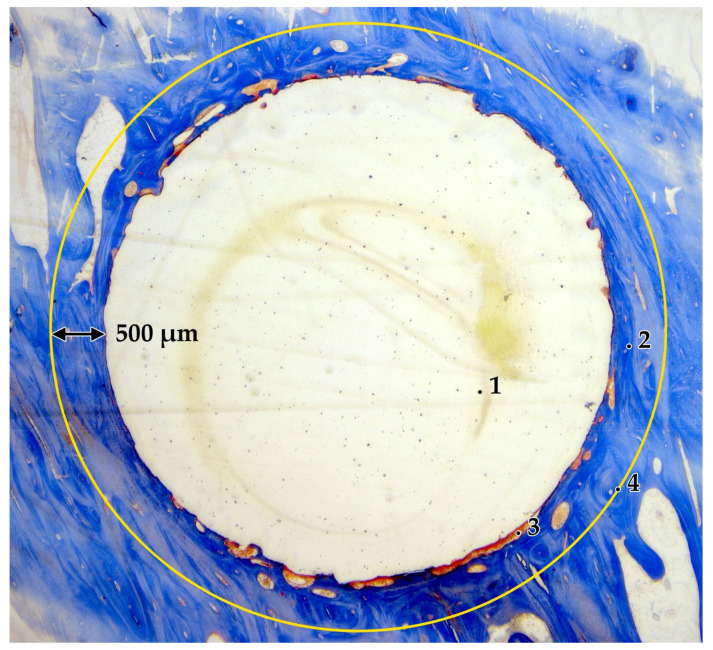
Masson-Goldner stained undecalcified section; (1: biomaterial, 2: bone, 3: soft tissue, 4: ROI 500 µm (yellow circle)).

**Figure 10 ijms-24-06200-f010:**
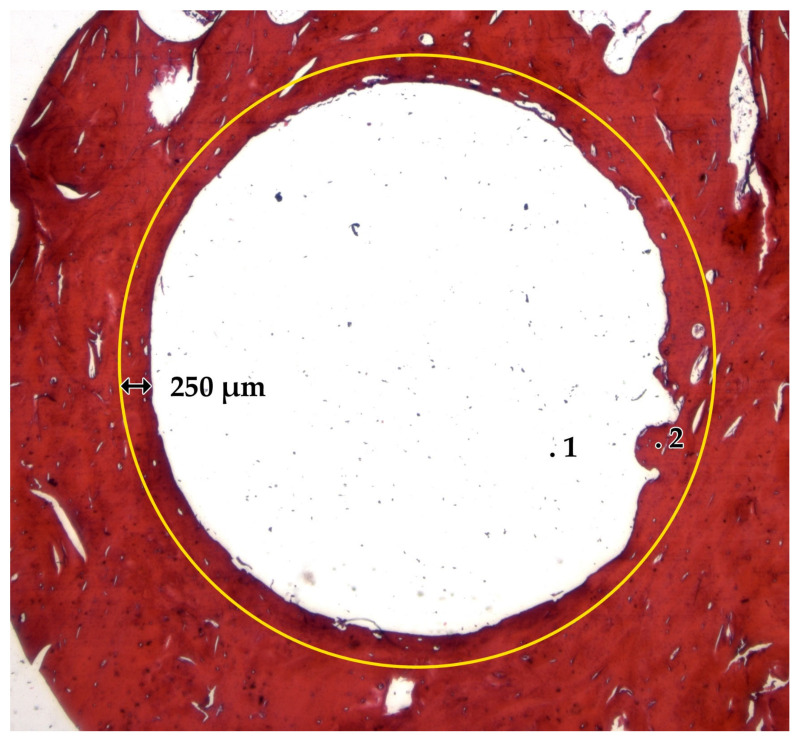
Decalcified and HE stained sections with a ROI of 250 µm (yellow circle). Biomaterial body (1) is in close contact to the surrounding bone (2).

**Figure 11 ijms-24-06200-f011:**
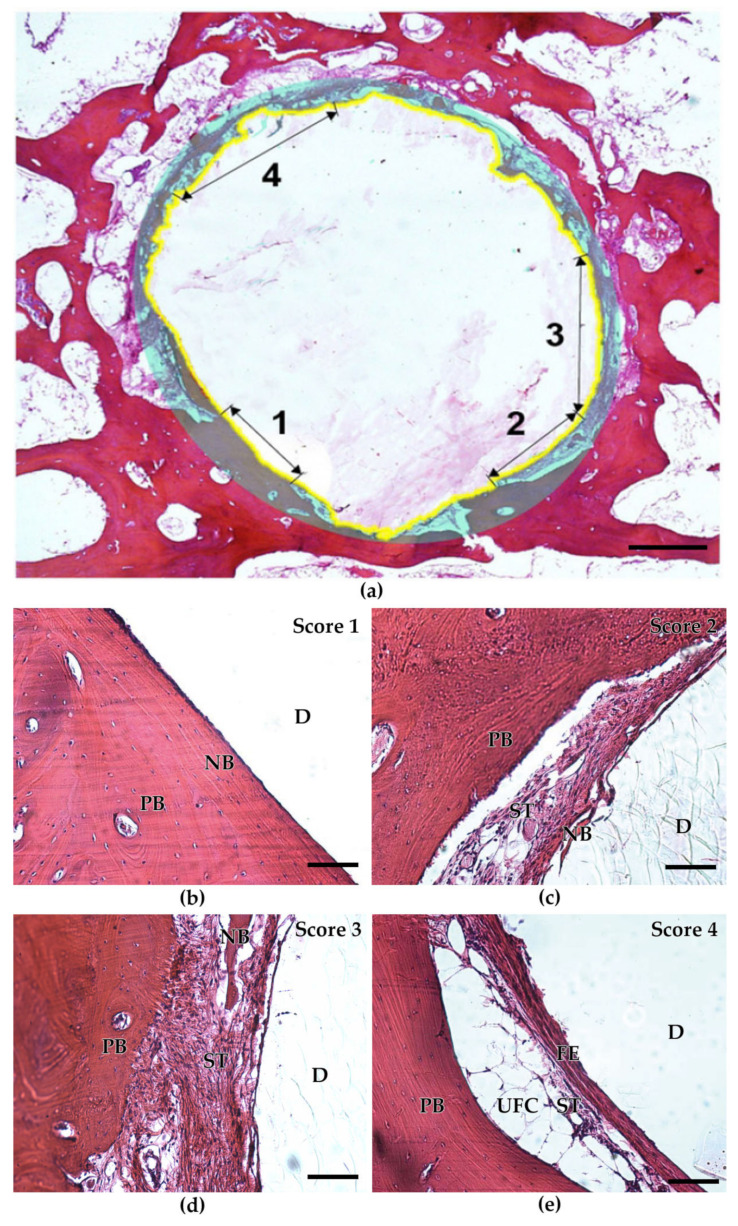
Four-graded histological evaluation score: (**a**) summary of all scores (1 to 4) blended in one single image. The ROI is marked by a greyish circular zone. Bone-to-biomaterial interface is indicated by a yellow line. Scale bar 500 µm; (**b**–**e**) respective examples for each evaluation score. Scale bar 100 µm. (D: defect zone with biomaterial remnants, NB: new-formed bone, ST: soft tissue, PB: peripheral bone, UFC: univacuolar fat cells, FE: fibrous encapsulation).

**Figure 12 ijms-24-06200-f012:**
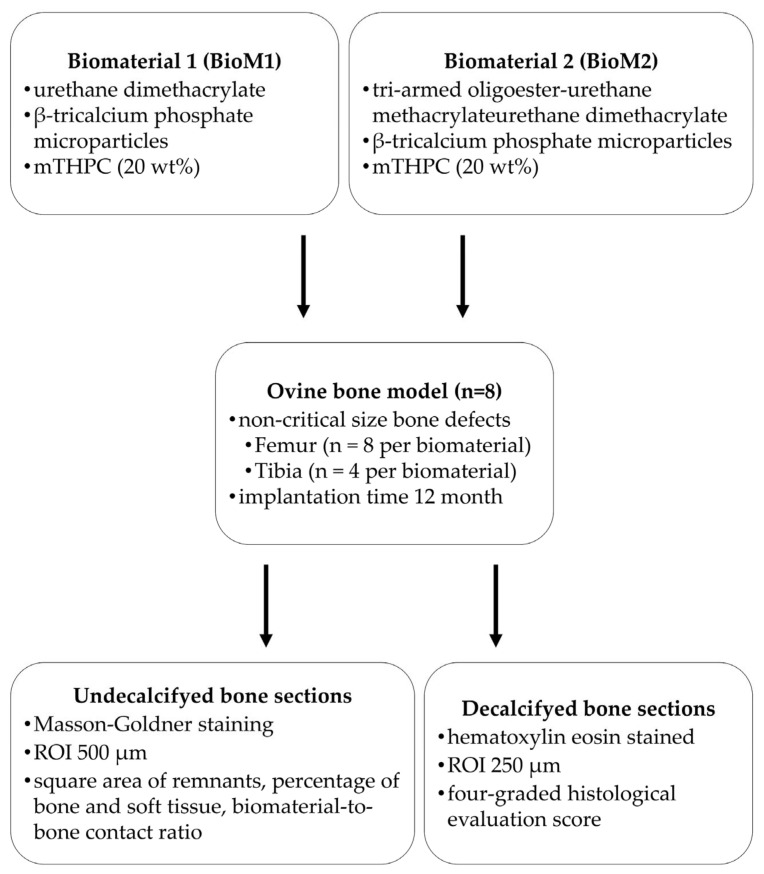
Workflow of the study.

**Table 1 ijms-24-06200-t001:** Location and total number of bone defects filled with BioM1 or BioM2.

	Animal ID	Sample ID	Femur	Tibia
BioM1	S4	S4FP	2	-
		S4TP	-	1
	S5	S5FP	2	-
		S5TP	-	1
	S6	S6FP	2	-
		S6TP	-	1
	S7	S7FP	2	-
		S7TP	-	1
			8	4
BioM2	S12	S12FP	2	-
		S12TP	-	1
	S13	S13FP	2	-
		S13TP	-	1
	S14	S14FP	2	-
		S14TP	-	1
	S15	S15FP	2	-
		S15TP	-	1
			8	4

**Table 2 ijms-24-06200-t002:** Four-graded histological evaluation score adapted and modified from DIN EN ISO 10993-6 [69].

Score	Definition
1	Completely mineralized bone with the presence of osteoblasts and/or osteocytes
2	Deposited connective tissue within the bone matrix
3	Connective tissue without signs of bone in the ROI
4	Additional appearance of univacular fat cells

## Data Availability

Available upon request from the corresponding author.

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
