# Peer review of "Histological and Histomorphometric Evaluation of Implanted Photodynamic Active Biomaterials for Periodontal Bone Regeneration in an Animal Study"

_ijms, 2023, doi:10.3390/ijms24076200_

Round 1

Reviewer 1 Report

Many thanks to the reviewer for the paper submission. This is a nicely written article, however some modifications are required in order to proceed to publication. The paper reports a promising use of BioM1 in periodontal regeneration.

1) in the introduction section at line 45 please modify the following phrase "...which also include the application of various grafting materials such as membranes, growth and differentiation factors and enamel matrix proteins.." 

into

"...with the use of resorbable or non resorbable membranes, growth and differentiation factors, enamel matrix proteins and autologous, heterologous and/or xenograft.."

please cite the following 

Chisci G, Fredianelli L. Therapeutic Efficacy of Bromelain in Alveolar Ridge Preservation. Antibiotics (Basel). 2022 Nov 3;11(11):1542. doi: 10.3390/antibiotics11111542. PMID: 36358197; PMCID: PMC9687015.

Jepsen S, Gennai S, Hirschfeld J, Kalemaj Z, Buti J, Graziani F. Regenerative surgical treatment of furcation defects: A systematic review and Bayesian network meta-analysis of randomized clinical trials. J Clin Periodontol. 2020 Jul;47 Suppl 22:352-374. doi: 10.1111/jcpe.13238. PMID: 31860125.

Paolantonio M, Di Tullio M, Giraudi M, Romano L, Secondi L, Paolantonio G, Graziani F, Pilloni A, De Ninis P, Femminella B. Periodontal regeneration by leukocyte and platelet-rich fibrin with autogenous bone graft versus enamel matrix derivative with autogenous bone graft in the treatment of periodontal intrabony defects: A randomized non-inferiority trial. J Periodontol. 2020 Dec;91(12):1595-1608. doi: 10.1002/JPER.19-0533. Epub 2020 Jun 17. PMID: 32294244.

Sassano P, Gennaro P, Chisci G, Gabriele G, Aboh IV, Mitro V, di Curzio P. Calvarial onlay graft and submental incision in treatment of atrophic edentulous mandibles: an approach to reduce postoperative complications. J Craniofac Surg. 2014;25(2):693-7. doi: 10.1097/SCS.0000000000000611. PMID: 24621726.

2) in the method section please specify the nature of this paper: retrospective or prospective, and indicate the methodology: number of cases, characteristics of the cases into a anagraphical table

3) please add a clinical image of the experimental procedure

4) please add a conclusion chapter

Author Response

Dear Reviewer,

we are very thankful for your comments that helped us to improve our manuscript. We tried to correct the article to the best of our knowledge. Changes among the manuscript are highlighted in green. Please find below the answers to your questions. Again, many thanks for your valuable time.

Sincerely yours,

Dr. Stefan Kranz

Comment:

In the introduction section at line 45 please modify the following phrase"...which also include the application of various grafting materials such as membranes, growth and differentiation factors and enamel matrix proteins.." into: "...with the use of resorbable or non resorbable membranes, growth and differentiation factors, enamel matrix proteins and autologous, heterologous and/or xenograft.."

Answer:

We modified the section as follows: “In those cases, different non-surgical and surgical regenerative procedures are applied which also involves the use of resorbable or non-resorbable membranes, growth and differentiation factors, enamel matrix proteins and autologous, heterologous bone grafts and/or xenografts [4-10].”. Please view lines 43 – 46.

Comment:

Please cite the following:

Chisci G, Fredianelli L. Therapeutic Efficacy of Bromelain in Alveolar Ridge Preservation. Antibiotics (Basel). 2022 Nov 3;11(11):1542. doi: 10.3390/antibiotics11111542. PMID: 36358197; PMCID: PMC9687015.

Jepsen S, Gennai S, Hirschfeld J, Kalemaj Z, Buti J, Graziani F. Regenerative surgical treatment of furcation defects: A systematic review and Bayesian network meta-analysis of randomized clinical trials. J Clin Periodontol. 2020 Jul;47 Suppl 22:352-374. doi: 10.1111/jcpe.13238. PMID: 31860125.

Paolantonio M, Di Tullio M, Giraudi M, Romano L, Secondi L, Paolantonio G, Graziani F, Pilloni A, De Ninis P, Femminella B. Periodontal regeneration by leukocyte and platelet-rich fibrin with autogenous bone graft versus enamel matrix derivative with autogenous bone graft in the treatment of periodontal intrabony defects: A randomized non-inferiority trial. J Periodontol. 2020 Dec;91(12):1595-1608. doi: 10.1002/JPER.19-0533. Epub 2020 Jun 17. PMID: 32294244.

Sassano P, Gennaro P, Chisci G, Gabriele G, Aboh IV, Mitro V, di Curzio P. Calvarial onlay graft and submental incision in treatment of atrophic edentulous mandibles: an approach to reduce postoperative complications. J Craniofac Surg. 2014;25(2):693-7. doi: 10.1097/SCS.0000000000000611. PMID: 24621726.

Answer:

We incorporated all references. Please view page 15, references 7-10.

Comment:

In the method section please specify the nature of this paper: retrospective or prospective, and indicate the methodology: number of cases, characteristics of the cases into an anagraphical table.

Answer:

We introduced the following sentence to the Materials and Methods section: “Eight female domestic sheep (Ovis gmelini aries) obtained from a local breeder with a mean age of 12 months were used in this prospective study.” Please view lines 103-104.

We also included table 1, which shows detailed information upon the used animals and collected bone specimen. Please view line 143.

Comment:

Please add a clinical image of the experimental procedure.

Answer:

We incorporated two new images. Please view figure 3 on page 4.

Comment:

Please add a conclusion chapter.

Answer:

A conclusion chapter was added. Please view lines 354-367.

Reviewer 2 Report

1.      The abstract should be broadened to give additional quantitative results.

2.      The present abstract was insufficient, please include the abstract's "take-home" message.

3.      Rearrange keywords alphabetically.

4.      I am encouraging the authors to not use abbreviations in the keywords.

5.      What is the novel bought by the authors in the current submission? Its works have been widely discussed in the past. Nothing something really new in the present form. The lack of a novel seems to make the present submission like to replication/modified work. The authors need to detail their novelty in the introduction section. It is a major concern for rejecting this paper.

6.      Previous studies must be explained in the introductory part, including their work, innovation, and limits, to demonstrate the research gaps that will be filled in the current literature.

7.      Please note that last paragraph of the introduction section of this article should be explained the present article objectively.

8.      Line 44-49, the auhtors explained biomaterials used for regeneration effort. But not discussed in depth than specifically directed to polymers in the end. The authors needs to discuss the other materials such as metals (magnesium). For this purpose, additional reference is needs to adopted as follows: Level of Activity Changes Increases the Fatigue Life of the Porous Magnesium Scaffold, as Observed in Dynamic Immersion Tests, over Time. Sustainability 2023, 15, 823. https://doi.org/10.3390/su15010823

9.      What is the limitation of the present work? Please include it before the conclusion section.

10.   Where in conclusion section? Provide it.

11.   In the conclusion section, further research must be discussed.

12.   In the whole of the manuscript, the authors sometimes made a paragraph only consisting of one or two sentences that made the explanation not clearly understood. The authors need to extend their explanation to become a more comprehensive paragraph. In one paragraph, it is recommended to consist of at least 3 sentences with 1 sentence as the main sentence and the other sentences as supporting sentences.

Author Response

Dear Reviewer,

we are very thankful for your comments that helped us to improve our manuscript. We tried to correct the article to the best of our knowledge. Changes among the manuscript are highlighted in yellow. Please find below the answers to your comments. The linguistic quality of this submission was also carefully reviewed and revised by an experienced native speaker. Again, many thanks for your valuable time.

Sincerely yours,

Dr. Stefan Kranz

Comment:

The abstract should be broadened to give additional quantitative results.

Answer:

The results section in the abstract was corrected and additional information introduced. Because the word count is limited to 200 words maximum, we decided to present only the most significant results in the abstract. Please view lines 20-28.

Comment:

The present abstract was insufficient, please include the abstract's "take-home" message.

Answer:

We reworked the abstract and included the following information: “Implantation of BioM2 (oligoester-urethane methacrylate-based) caused accumulation of lymphocytes and giant cells and led to fibrous encapsulation.” […] “BioM1 (urethane dimethacrylate-based) showed favorable regenerative characteristics and can be recommended for further studies. Please view lines 25-28.

Comment:

Rearrange keywords alphabetically.

Answer:

All keywords were arranged in alphabetical order.

Comment:

I am encouraging the authors to not use abbreviations in the keywords.

Answer:

There are no more abbreviations in the keywords.

Comment:

What is the novel bought by the authors in the current submission? Its works have been widely discussed in the past. Nothing something really new in the present form. The lack of a novel seems to make the present submission like to replication/modified work. The authors need to detail their novelty in the introduction section. It is a major concern for rejecting this paper.

Answer:

In the submitted manuscript biocompatibility and osseointegration of two different synthetic polymers with photodynamic antimicrobial surface activity were observed. Both materials were developed for regeneration of persistent periodontal intrabony defects. Up to now, none of the currently available materials meet all necessary clinical requirements such as providing sufficient mechanical stability paired with efficient osteoinductive and -conductive properties as well as mechanisms to fight local infections of the implant site. To fill this gap, our group developed these two photosensitizer-armed materials. Our group already published data showing that both biomaterials are of promising characteristics in regard to their mechanical stability/ resistance, cytocompatibility and antimicrobial activity. Information upon the biocompatibility and performance in real bone defects are hence missing, but are provided by the submitted manuscript. As requested, we introduced further information upon these issues to the manuscript. For detailed information please view lines 50-56, 68-69 and 76-82.   

Comment:

Previous studies must be explained in the introductory part, including their work, innovation, and limits, to demonstrate the research gaps that will be filled in the current literature.

Answer:

Previous studies that focus on bone regeneration, especially in the periodontal region are applied. The major disadvantages and limitations of grafting materials currently available are explained within lines 50-53. The most important research of our own group was included. Please view references 16, 33-35 and 54-58. The aim of the study was addressed and information to fill the current information gap are provided. Please view lines 14-15, 25-28, 68-71 and 76-82.

Comment:

Please note that last paragraph of the introduction section of this article should be explained the present article objectively.

Answer:

The last paragraph of the introduction was reworked and additional new information included. The objective of the investigation is now provided. Please view lines 76-82.

Comment:

Line 44-49, the auhtors explained biomaterials used for regeneration effort. But not discussed in depth than specifically directed to polymers in the end. The authors needs to discuss the other materials such as metals (magnesium). For this purpose, additional reference is needs to adopted as follows: Level of Activity Changes Increases the Fatigue Life of the Porous Magnesium Scaffold, as Observed in Dynamic Immersion Tests, over Time. Sustainability 2023, 15, 823. https://doi.org/10.3390/su15010823

Answer:

We are very thankful for this valuable comment. Additional information was introduced. Please refer to the following paragraphs (lines 43-49): “After initial anti-infectious therapy, periodontal pockets might still persist which is often associated with the presence of deep intrabony defects [3]. 

In those cases, different surgical and non-surgical regenerative procedures are ap-plied which also involves the use of resorbable or non-resorbable membranes, growth and differentiation factors, enamel matrix proteins and autologous, heterologous bone grafts and/or xenografts [4-10].

Because of the high practicability, alloplastic bone grafts such as bioceramics (e.g. absorbable/non-resorbable hydroxyapatite, beta-tricalcium phosphate), bioglasses, metals, calcium phosphate cements and polymers are also of particular interest [11-15].”

Reference: “Level of Activity Changes Increases the Fatigue Life of the Porous Magnesium Scaffold, as Observed in Dynamic Immersion Tests, over Time. Sustainability 2023, 15, 823. https://doi.org/10.3390/su15010823” was included as reference # 15.

Comment:

What is the limitation of the present work? Please include it before the conclusion section.

Answer:

Limitations of the study are included. Please refer to lines 342-348.

342-348

Comment:

Where in conclusion section? Provide it.

Answer:

A conclusion section was introduced. Please view lines 354-367.

Comment:

In the conclusion section, further research must be discussed.

Answer:

Further research approaches such as regeneration studies with an additional need for structural support and investigations that focus upon the antimicrobial photodynamic activity in-vivo, especially in case of periodontal bone lesions are mentioned.

Comment:

In the whole of the manuscript, the authors sometimes made a paragraph only consisting of one or two sentences that made the explanation not clearly understood. The authors need to extend their explanation to become a more comprehensive paragraph. In one paragraph, it is recommended to consist of at least 3 sentences with 1 sentence as the main sentence and the other sentences as supporting sentences.

Answer:

Thank you for these recommendations. We checked the entire manuscript for short paragraphs and corrected them as requested.

Round 2

Reviewer 1 Report

Accept

Author Response

Dear reviewer,

many thanks for your valuable time.

Best regards,

Dr. Stefan Kranz

Reviewer 2 Report

1.      Recommended to the authors provide an additional figure in the introduction section with related submissions after revision to improve the article presentation.

2.      Instead of only using the dominating text as a present form, the authors should also include extra illustrations in the form of figures that clarify the workflow of the current study to make the reader's understanding simpler.

3.      Why not used human in the present study? Any basis?

4.      Manufacturer, country, and specification information for experimental setup should be presented with more specificity.

5.      Error and tolerance of experimental tools used in this work are important information that needs to be explained in the manuscript. It is would use as a valuable discussion due to different results in the further study by other researcher.

6.      Explain the importance parameter of scaffold, there are porosity. Also, is the author consider tortuosity as the parameter? If not, explain it as study’s limitation. Refer the relevant ference for this purpose as follows: The Effect of Tortuosity on Permeability of Porous Scaffold. Biomedicines 2023, 11, 427. https://doi.org/10.3390/biomedicines11020427

7.      A comparison of the results with similar past investigations is required.

8.      Please, that major improvement is needed in the discussion section of the present manuscript, where the present form is not enough.

9.      Literature from the last five years should be enriched to reference. MDPI reference is strongly recommended.

10.   I suggest to the authors for reducing the number of literature from self-citation.

11.   English needs to be proofread due to grammatical errors and English style.

12.   It is suggested to the authors for providing graphical abstract in the system after revision.

Author Response

Dear Reviewer,

thank you for your comments that helped us to improve our manuscript. All new changes were highlighted in yellow. We tried to correct each point to the best of our knowledge. Please find below the answers to your remarks.

Thank you again for your valuable time!

Sincerely yours,

Dr. Stefan Kranz

Comment:

Recommended to the authors provide an additional figure in the introduction section with related submissions after revision to improve the article presentation.

Answer:

In our opinion it is not necessary to include an additional figure to the introduction section that summarizes current literature. The present manuscript was not designed of being a review article. In the introduction the study is briefly placed in the context of the current scientific point of view. Also, it is explained why the study is important. If references are still missing, we kindly ask the reviewer to send a list of important literature that still needs to be included into the introduction section.

Comment:  

Instead of only using the dominating text as a present form, the authors should also include extra illustrations in the form of figures that clarify the workflow of the current study to make the reader's understanding simpler.

Answer:

An additional figure that shows the workflow was introduced to the manuscript. Please view figure 8.

Comment:

Why not used human in the present study? Any basis?

Answer:

We do not understand the reviewers` intention here. Biocompatibility and osteointegration of BioM1 and BioM2 were not yet investigated in-vivo. Therefore, animal studies are mandatory.

Comment:

Manufacturer, country, and specification information for experimental setup should be presented with more specificity.

Answer:

We checked all specifications for inconsistency. However, should additional information still be required, we kindly ask the reviewer to point towards the specification he/she would like to have changed.

Comment:

Error and tolerance of experimental tools used in this work are important information that needs to be explained in the manuscript. It is would use as a valuable discussion due to different results in the further study by other researcher.

Answer:

All methods described in the present submission are standardized procedures within our facility. We did not face any limitations or errors of the applied experimental tools. The methods are described in detail and are also reproducible. 

Comment:

Explain the importance parameter of scaffold, there are porosity. Also, is the author consider tortuosity as the parameter? If not, explain it as study’s limitation. Refer the relevant ference for this purpose as follows: The Effect of Tortuosity on Permeability of Porous Scaffold. Biomedicines 2023, 11, 427. https://doi.org/10.3390/biomedicines11020427

Answer:

Thank you for this important comment. We included further information upon the issue. The listed reference was also included (53). Please refer to lines 315-318 and 353-355.

Comment:

A comparison of the results with similar past investigations is required.

Answer:

Thank you for this comment. We introduced additional information to the discussion section. Please refer to lines 277 to 288.

Comment:

Please, that major improvement is needed in the discussion section of the present manuscript, where the present form is not enough.

Answer:

In the present study osseointegration and biocompatibility of two polymeric materials with photodynamic antimicrobial activity was investigated in an ovine bone defect model. To the best of our knowledge there are currently no examinations that observe the performance of photodynamic active polymeric alloplasts in-vivo. Furthermore, photodynamic active grafting materials for the treatment of infected oral/periodontal bone lesions are currently not yet introduced. The mentioned points are already included in the present study. As requested, additional information for increasing the significance of the submission was introduced. Please refer to lines 292-300.

Comment:

Literature from the last five years should be enriched to reference. MDPI reference is strongly recommended.

Answer:

The list of references was updated. Twenty-seven mdpi-based references are now included.

Comment:

I suggest to the authors for reducing the number of literature from self-citation.

Answer:

We deleted some of the references. This point is contradictory, since the reviewer emphasized to rise the number of own references during the first revision: “[…] previous studies must be explained in the introductory part, including their work […].”

Comment:

English needs to be proofread due to grammatical errors and English style.

Answer:

The linguistic quality of the manuscript was carefully checked by a native speaker from Ohio (USA), who is a linguist holding three academic degrees. For this reason, we cannot follow the repeated criticism of reviewer #2 regarding "grammatical errors and English style". Therefore, we would like to ask the reviewer to get more constructive in this point by providing examples of what he/she thinks needs to be improved linguistically.

Comment:

It is suggested to the authors for providing graphical abstract in the system after revision.

Answer:

A graphical abstract has been included.

Round 3

Reviewer 2 Report

Well done to the authors, I have one suggestion to the authors. Since the present study performs in vivo, potential further study adopting in silico (computational simulation) needs to explained since it bring several advantages such as lower cost and faster results compared to in vivo. For this purpose, refer the relevant reference as follows: Adopted Walking Condition for Computational Simulation Approach on Bearing of Hip Joint Prosthesis: Review over the Past 30 Years. Heliyon 2022, 8, e12050. https://doi.org/10.1016/j.heliyon.2022.e12050

Author Response

Dear Reviewer,

thank you for your valuable comment. We introduced another paragraph to the discussion section about computational simulation. The mentioned reference has been additionally included (48). Please also view lines 301-306.  All new changes were highlighted in yellow.

Thank you again for your valuable time and your help in preparing our manuscript.

Sincerely yours,

Dr. Stefan Kranz